# COPD: The Annual Cost-Of-Illness during the Last Two Decades in Italy, and Its Mortality Predictivity Power

**DOI:** 10.3390/healthcare7010035

**Published:** 2019-03-01

**Authors:** Roberto W. Dal Negro

**Affiliations:** National Centre for Respiratory Pharmacoeconomics and Pharmacoepidemiology, 37124 Verona, Italy; robertodalnegro@gmail.com

**Keywords:** COPD, cost-of-illness, economic outcomes, PRO-BODE Index, prediction of mortality

## Abstract

Chronic obstructive pulmonary disease (COPD) is a progressive pathological condition characterized by a huge epidemiological and socioeconomic impact worldwide. In Italy, the actual annual cost of COPD was assessed for the first time in 2002: the mean cost per patient per year was €1801 and ranged from €1500 to €3912, depending on COPD severity. In 2008, the mean annual cost per patient was €2723.7, ranging from €1830.6 in mild COPD up to €5451.7 in severe COPD. In 2015, it was €3291, which is 20.8% and 82.7% higher compared to the costs estimated in 2008 and 2002, respectively. In all these studies, the major cost component was direct costs, in particular hospitalization costs due to exacerbations, which corresponded to 59.9% of the total cost and 67.2% of direct costs, respectively. When the annual healthcare expenditure per patient is related to the length of survival by means of the PRO-BODE Index (PBI, which is the implementation of the well-known BODE Index with costs due to annual exacerbations and/or hospitalizations), the annual cost of care proved much more strictly and inversely proportional to patients’ survival at three years, with the highest regression coefficient (r = −0.58) of all the multidimensional indices presently available, including the BODE Index (r = −021). In Italy, even though tobacco smoking has progressively declined by up to 21% in the general population, the economic impact of COPD has shown relentless progression over the last two decades, confirming that the present national health system organization is still insufficient for facing the issue of chronic diseases, in particular COPD, effectively. The periodic assessment of costs is an effective instrument for care providers in predicting COPD mortality, and for decision makers for updating and planning their social, economic, and political strategies.

## 1. Background

Chronic obstructive pulmonary disease (COPD) is a complex and progressive condition which is characterized by a dramatic socioeconomic impact worldwide [1].

The epidemiological, clinical, and socioeconomic impacts of COPD are still constantly increasing, and COPD is projected to be the third leading cause of death in the world by 2030, and the seventh highest in terms of the burden of disease [2]. On the other hand, people’s mean age is progressively rising, and a further, progressive increase in the prevalence and the incidence of chronic diseases, including COPD, is therefore expected.

The growing interest in pharmacoeconomic issues reflects the ever-growing need for “accountability” and for assessing the economic value of health strategies oriented towards optimization of healthcare resource allocation.

Even though health economic data are not easily comparable among different countries due to the differences that exist in their national health systems, a common critical point is represented by the high overall burden of COPD, with the increase in costs proportional to its clinical severity. The major cost components mainly depend on the number and severity of COPD exacerbations, the hospitalization rates, the high proportion of costs related to acquired severe disability, the need for long-term oxygen therapy (LTOT), and insufficient coverage for drug expenses [3].

The periodic checking of COPD resource consumption represents a crucial indicator for assessing the impact of COPD on the overall health system and the community in all countries. The assessment of the economic burden of chronic disease is constantly on the agenda of healthcare policy makers as they have to face the ever-growing need to reconcile the limited availability of economic resources with the constant addition of new therapeutic options in all healthcare areas, and in aging populations. The economic crisis has worsened this context, with national healthcare budgets generally shrinking in European countries.

From a general point of view, a key assumption is that chronic diseases, in particular COPD, negatively affect not only patients’ lives, but also those of their relatives and caretakers, thus representing a burden for society as a whole. Patients experience suffering, inactivity, limitations, and invalidity that cause a worsening in their Quality of Life (QoL) and could lead to disability and to premature death in some cases. As a consequence, patients’ families undergo disruption as well as emotional and financial hardship. Society, as a whole, suffers from the economic burden of COPD, in terms of morbidity, days lost from work, early retirement, and premature death [4].

The “era of accountability” started around the 1980s. The Cost of Illness (COI) method was initially established by Rice et al. [5] and carried out by means of the measurement of resource consumption and estimation of associated costs. It is a useful methodological tool widely accepted as a means of describing the economic burden of a given disease.

The systematic use of real world evidence data is regarded as a key issue in the development of credible economic analyses that can be used by institutions for future planning. Therefore, data that allow the periodical estimation and updating of such costs should ideally be produced in a real clinical context, particularly in Italy where the governance and management of COPD is still suboptimal.

## 2. Cost Trends in Real Life

In Italy, the COPD cost-of-illness was assessed in real life for the first time in 2002, by means of a multicenter study carried out within the Tri-Veneto Region (including Veneto, Trentino-Alto Adige, and Friuli-Venezia Giulia) in Italy, corresponding to 5.5 million inhabitants. The mean cost per patient per year was €1801, ranging from €1500 to €3912, and was dependent on COPD severity and on the presence of comorbidities [6].

Data showed that the major components of cost were direct costs related to exacerbations and hospitalizations, while costs due to pharmacological and nonpharmacological treatments were much lower. Data from this study emphasized that COPD was often misdiagnosed (namely, the mean time interval between the first GP visit due to symptoms and the first lung function test was 6 ± 5 years) and mistreated, and that COPD management was less effective and highly expensive, particularly in patients with the highest degrees of severity.

In 2008, the Social Impact of Respiratory Integrated Outcomes (SIRIO) study provided the first estimate of the economic burden of COPD in Italy. The SIRIO study was designed as a multicenter global outcome study with the aim of producing data regarding the socioeconomic impact of COPD, before and one-year following treatment optimization (according to current Global Outcomes in Lung Disease (GOLD) guidelines) [7].

The study was carried out in 37 specialist centers evenly distributed throughout the country. At baseline, the severity of COPD, graded according to current GOLD guidelines, was 24.2% mild, 53.7% moderate, and 16.8% severe COPD. At baseline, spirometry had already been performed in 64% of patients.

The mean annual cost per patient at the first visit was €2723.7 and it increased proportionally with COPD severity, ranging from €1314.9 (±1830.6 sd) in mild to €5451.7 (±5312.7 sd) in severe COPD. After the end of the twelve-month follow-up, the mean cost per patient decreased by €590.8 compared to baseline. The total average direct costs per patient decreased from €2506.8 in the previous year to €2044.6 at the end of follow-up, corresponding to a decrease of €462.3. We note that in spite of an increase of €361.5 in costs for pharmaceutical treatments, there was nevertheless a significant decrease in all other direct costs: in particular, hospitalization and emergency care costs dropped by €718.6 and indirect mean costs decreased by €128.5. However, as mentioned above, the increase in costs of basic therapy was offset by a marked, systematic decrease in all other direct costs, as well as in the indirect costs induced by the disease.

The last study that aimed to update the cost of COPD in Italy was conducted in 2015 [8].

Direct cost accounted for 89.1% of the total cost. The total mean annual cost per patient was €3291, 20.8% higher than that estimated in 2008 (€2724) in a similar (in terms of age, gender distribution, and disease severity) cohort of patients [7]. In particular, the figures of different components of cost were hospitalization €1970, outpatient care €463, pharmaceutical €498.6, and indirect costs €358. The total cost of COPD accounted for 0.80 points of the National Gross Product in 2015.

In this study, the hospitalization cost corresponded to 59.9% of the total cost and 67.2% of the direct cost. These ratios were very close to those found in 2008 (59.4% and 64.5%, respectively). Moreover, when compared to 2008 [7], the outpatients’ cost increased by 29.9%, changing from €356.7 six years ago to the present value of €463.2, while that of the pharmacological treatment increased by 43.6%, changing from €347.2 to the present value of €498.6. This value corresponds to 17.0% of the direct cost and to 15.1% of the total cost of illness, respectively.

The mean annual cost per patient dropped significantly after the twelve-month follow-up from €3290.7 to €2706.7, which corresponded to a saving of €584.0 (17.7% reduction) (*p* < 0.0001). The direct cost decreased by 16.1%, corresponding to a saving of €471.8 (*p* < 0.0001), while indirect costs dropped by 31.3%, corresponding to a saving of €112.2 (*p* < 0.001). At the end of the follow-up, direct costs represented 57.9% of the total cost and 63.8% of direct costs, respectively, corresponding to 3–4% less than observed at the first visit. Outpatient costs substantially decreased by 25.8%: from €463.2 to €343.9 (*p* < 0.0001). Different from other cost categories, pharmaceutical costs increased during the follow-up, even if not in a significant manner (*p* = ns). The expense for drugs changed by 9.7% from €498.6 to €546.8. No significant difference was reported in any cost category between genders (all *p* = ns), even though a general tendency to lower costs was seen in females for all costs, except for pharmaceutical costs [8].

Even if at a smaller extent than in 2002 and 2008 [6,7], the results of the study carried out in 2015 once again confirmed that the COPD cost-of-illness is still increasing substantially (Figure 1), and that there is still a clear tendency to manage COPD in the hospital setting in Italy, independent of its intrinsic severity. The most recent breakdown of COPD cost in Italy is reported in Table 1.

In the last decade, multidimensional scores have been introduced in order to better assess COPD outcomes, particularly the mortality risk, even if results were variable due to their different specificity and sensitivity [3,9,10,11,12,13,14,15,16,17,18,19,20,21,22,23]. The vast majority of studies have stated that exacerbations represent the main driver of COPD burden because they can affect morbidity, quality of life, hospitalizations, mortality, and related healthcare expenditure [24,25,26,27,28].

From a general point of view, the use of multidimensional grading systems improved the sensitivity of the mortality risk assessment in COPD patients as these instruments valued several factors affecting severity and prognosis of COPD, albeit to different levels of specificity.

In fact, all multidimensional indices are primarily oriented to a clinical approach, but fail to take into account the utilization of healthcare resources caused by exacerbations and/or hospitalizations. In the past, the consumption of healthcare resources was never considered as a major index for assessing the impact of COPD even though the largest proportion of the annual economic impact proved to be related to the exacerbation and hospitalization events occurring annually in COPD, which, on the other hand, can also affect the length of survival.

Quite recently, the annual cost-of-illness has been used for the first time as a predictor of mortality in a sample of 275 COPD patients of different severities surveyed for three years [29].

The annual cost-of-illness was then assessed by stratifying the overall cohort by the ultimate outcome (i.e., survival or death) and implemented to the corresponding BODE index (such as the most specific of the previous indices for predicting mortality) during a three-year observation period (Figure 2). The BODE Index, which consists of four components (namely, the body mass index (BMI), the post-bronchodilator FEV1 in % predicted, the dyspnea score measured by the modified MRC scale, and the 6-min walk distance in meters) [3], has been accepted within the scientific community since 2004 as the most specific of all indices usable for predicting mortality.

The novel index thus emerging from this study was named the PRO-BODE index, and it was graded according to a specific algorithm, ranging from 0–10 points, as the BODE Index was. When compared to usual indices (such as clinical symptoms, the complete lung function, the components of the BODE index, the Charlson Comorbidity Index, the exacerbation rate, and the hospitalization rate) and to other multidimensional indices presently available (namely, BODE, m-BODE, e-BODE, BODE-x, ADO, DOSE, COPD Prognostic, and SAFE), the PRO-BODE score proved much more strictly proportional to the cost of care and inversely proportional to the patients’ survival at three years. In particular, the PRO-BODE regression power was the strongest (r = −0.58) and also much higher than that of the BODE index (r = −0.21), which was previously the best predictor of mortality [29].

Furthermore, the progression rate of costs assessed by the PRO-BODE score followed a geometrical trend [29] (Figure 3). In other words, the higher the PRO-BODE score (which takes into account the two major components of the annual cost-of-illness), the shorter the survival at three years, and the higher the predictivity power of the novel PRO-BODE in terms of COPD survival (Table 2).

## 3. Conclusions

The search for better socioeconomic conditions led Western Countries to uncontrolled economic development in the last century, regardless of the occurrence of future socioeconomic effects, such as the dramatic increase in air pollution due to industrial emissions, the never-ending increase of energy requirements, the superconcentration of people in hypercrowded towns, the great increase in vehicular traffic, the increase of smoking habit, and the dramatic changes in lifestyle. 

COPD is a slowly progressive pathological condition which was confirmed to represent one of the major causes of chronic morbidity and mortality worldwide, such as the third leading cause of death in the world by 2030 and the seventh in terms of burden of disease [4,5].

Unfortunately, in Italy, COPD is also still poorly or insufficiently perceived by both the general population and the majority of decision makers [30], in terms of long-term clinical, social, and economic consequences.

COPD impact has gradually increased over the last two decades in Italy [6,7,8].

Effective actions are urgently needed in terms of prevention, pharmacological and nonpharmacological treatments, therapeutic education, increasing smoking cessation campaigns, health information, and specific health planning with the aim of containing the clinical and economic burdens of COPD.

At present, the economic resources consumed for outpatient care are higher than in the past, but they are still insufficient for changing attitudes towards the all-too-frequent hospitalizations. In fact, it depends on the present organization of the Italian health system, which is still insufficient for facing the issue of chronic diseases and also COPD, independent of clinical severity. A substantial decrease in the economic impact of COPD will likely occur only when the public and the decision makers’ awareness increases substantially, when specific plans are strategically implemented all over Italy, and when GPs and territorial lung physicians are in the cultural and operational conditions to efficiently manage this type of patient.

Finally, the periodic assessment of the real-life cost of COPD represents the most effective instrument in the hands of healthcare providers for predicting mortality and in the hands of decision makers for updating, planning, and checking their interventional strategies.

## Figures and Tables

**Figure 1 healthcare-07-00035-f001:**
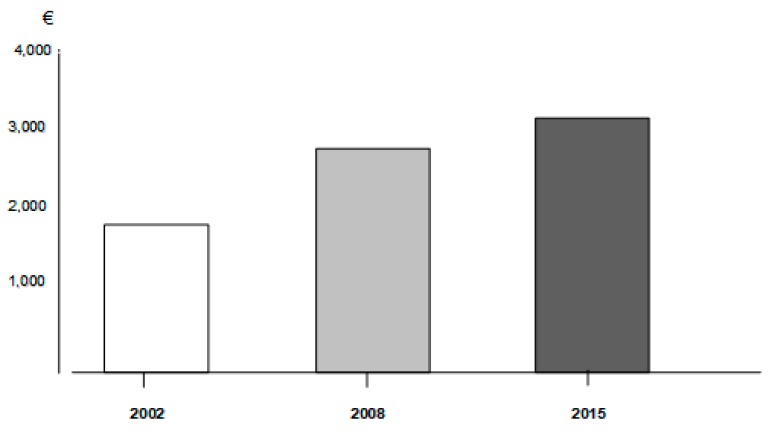
The trend of the annual chronic obstructive pulmonary disease (COPD) cost-of-illness from 2002 to 2015 in Italy.

**Figure 2 healthcare-07-00035-f002:**
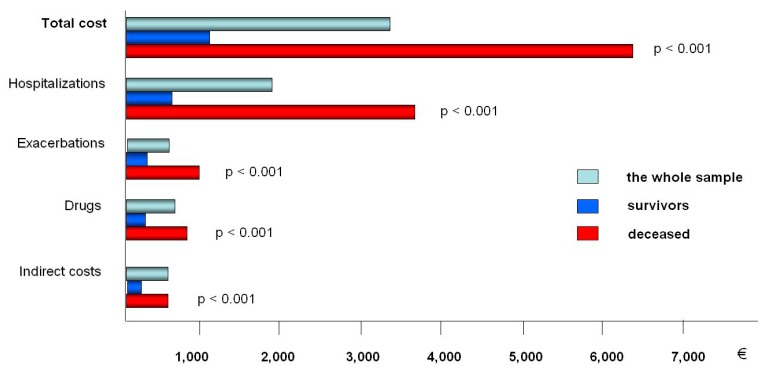
The mean total annual cost per patient and the corresponding components of cost (due to hospitalizations, nonadmitted exacerbations, pharmacological treatment, and indirect costs) in the whole sample, in survivors at three years and in those patients who died during the three years of survey.

**Figure 3 healthcare-07-00035-f003:**
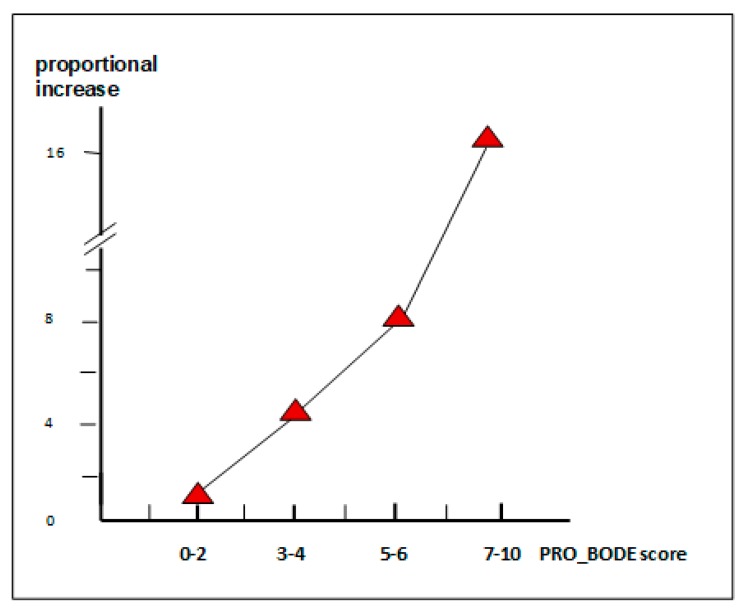
The increasing rate of costs by the PRO-BODE score.

**Table 1 healthcare-07-00035-t001:** The last breakdown of chronic obstructive pulmonary disease (COPD) cost-of-illness in Italy in 2015 [8].

Classification	Costs (Mean (95% CI))
Direct Costs	2932.2 (2643.1, 3221.3)
Hospitalization Costs	1970.4 (968.0, 2972.8)
Out-patient Costs	463.2 (207.5, 718.9)
Pharmaceutical Costs	498.6 (252.5, 744.7)
Indirect Costs	358.5 (119.0, 598.0)
Total Costs	3290.7 (2539.9, 4051.2)

CI: confidence interval.

**Table 2 healthcare-07-00035-t002:** The inverse relation between the cost of COPD and the patients’ survival according to the PRO-BODE score.

Pro-Bode Score (Points)	Survival (Days)	Cost (€)
0–2 (*n* = 142)	1023.8 (198.9)	494.8 (1,454.2)
3–4 (*n* = 66)	889.5 (239.4)	2040.9 (2079.0)
5–6 (*n* = 36)	762.2 (283.4)	4952.9 (2265.3)
7–10 (*n* = 31)	752.1 (226.7)	9224.9 (7804.2)

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
