# Peer review of "COPD: The Annual Cost-Of-Illness during the Last Two Decades in Italy, and Its Mortality Predictivity Power"

_healthcare, 2019, doi:10.3390/healthcare7010035_

Round 1

Reviewer 1 Report

Major comments:

1) It’s very confusing to mix methods and results in the same section “the trend of costs in real life”. Please reference manuscript submission overview here https://www.mdpi.com/journal/healthcare/instructions and edit the manuscript properly.

2) Do you have more information on COPD cost-of-illness other than 2002, 2008, and 2015? Are there reasons you only examined three years?

3) As PRO-BODE index is an important concept in this paper, the author should describe this index more in details either in introduction or method part e.g. how to define this score

4)  The introduction lacks of describing a “gap” why this research is important to be done in Italy

Minor comments: I would suggest to review the manuscript thoroughly and check all grammatically incorrect sentences.

1)  “2.723,7,” seems like a type-O in the abstract and page 2.

2) Another type-O “The total average direct costs per patient decreased from 89 € 2,506.8 in the previous year to €2,044.6 at the end of follow-up, such as a drop by €462.3).”

3) It’s hard to understand what this sentence means especially the bold part. “In 2015, it was € 3,291, 13 such as a value higher by 20.8% and by 82.7% when compared to those estimated in 2008 and 2002, respectively.” In the abstract. It should be rephrased e.g. In 2015, it was € 3,291, 13 which is 20.8% and 82.7% higher values compared to those estimated in 2008 and 2002, respectively.  

4) No need to capitalize Countries across the manuscript.

5) Again, why the bold part is capitalized? To note 90 that, in spite of an increase of €361.5 in costs for pharmaceutical treatments, there was nevertheless a 91 significant decrease in all other direct costs: in particular, cost for hospitalization Day Hospital and 92 Emergency Care dropped by €718.6. 

6) Aim to “update”. “The last study aimed to updating the cost of COPD in Italy was conducted in 2015” 

7) Figure 1 has a gap between x and y axis.  

8) Table 1. Ref 8 should be cited the same way as other references. Also, is it mean and standard errors or simply min/max range? Is it referring 95% confidence intervals? The table is very confusing.  

9) Please properly label y axis in Figure 2 and 3. Figure 2 also has gaps between y axis and bar charts.

Author Response

Major comments:

1) It’s very confusing to mix methods and results in the same section “the trend of costs in real life”. Please reference manuscript submission overview here https://www.mdpi.com/journal/healthcare/instructions and edit the manuscript properly.

Response: As originally required by the Journal, the paper consists in a global review on  the trend of costs over the last two decades in Italy. Therefore, it was not divided in Materials, Methods, and Results as usually occurring in experimental papers: this, just in order to provide a conituous trend of information and of progressive changes in COPD cost.

2) Do you have more information on COPD cost-of-illness other than 2002, 2008, and 2015? Are there reasons you only examined three years?

Response: Specific data on COPD cost were not available before 2002 in Italy, as well as after 2015.

After 2002 (the first Italian study), the CESFAR National Centre planned to updating COPD costs every 6 years, while a 6-year interval between the studies was presumed to be long  enough for potentially assessing significant changes.

3) As PRO-BODE index is an important concept in this paper, the author should describe this index more in details either in introduction or method part e.g. how to define this score

Response: According to your sugegstion, information concerning the PRO_BODE Index had been implemented in the text.

4)  The introduction lacks of describing a “gap” why this research is important to be done in Italy

Response: The general relevance of periodic phamaco-economic studies had been suggested in lines 45-51 and 59-65 of the Introduction/Background.  Anyway, according to your suggestion, a specific mention concerning the importance of this kind of studies in Italy was added to the text.

Minor comments: I would suggest to review the manuscript thoroughly and check all grammatically incorrect sentences.

1)  “2.723,7,” seems like a type-O in the abstract and page 2.

Response: Reworded.

2) Another type-O “The total average direct costs per patient decreased from 89 € 2,506.8 in the previous year to €2,044.6 at the end of follow-up, such as a drop by €462.3).”

Response: Reworded.

3) It’s hard to understand what this sentence means especially the bold part. “In 2015, it was € 3,291, 13 such as a value higher by 20.8% and by 82.7% when compared to those estimated in 2008 and 2002, respectively.” In the abstract. It should be rephrased e.g. In 2015, it was € 3,291, 13 which is 20.8% and 82.7% higher values compared to those estimated in 2008 and 2002, respectively.  

Response: Reworded in the Abstract and in the Text according to your suggestion.

4) No need to capitalize Countries across the manuscript.

Response: Modified.

5) Again, why the bold part is capitalized? To note 90 that, in spite of an increase of €361.5 in costs for pharmaceutical treatments, there was nevertheless a 91 significant decrease in all other direct costs: in particular, cost for hospitalization Day Hospital and 92 Emergency Care dropped by €718.6. 

Response: Modified.

6) Aim to “update”. “The last study aimed to updating the cost of COPD in Italy was conducted in 2015” 

Response: Modified.

7) Figure 1 has a gap between x and y axis.  

Response: The x-y gap has been eliminated.

8) Table 1. Ref 8 should be cited the same way as other references. Also, is it mean and standard errors or simply min/max range? Is it referring 95% confidence intervals? The table is very confusing.  

Response: Even if the 95% CI had already been indicated at the bottom of tab.1, the corresponding caption has been implemented accordingly.

9) Please properly label y axis in Figure 2 and 3. Figure 2 also has gaps between y axis and bar charts.

Response: The y label  has been added infig. 2 (the cooresponding caption was already clear), while the caption of fig. 3 has been slightly reworded and clarified. No gaps are present in my version of figures 2 and 3...

Reviewer 2 Report

This is a non-medical purely economic assessment of COPD in Italy.

It provides no reference to the reasons for the increase in medical costs or to the trends in tobacco smoking that underlie the occurrence and prognosis of the disease.

It does not consider the change in prognosis that occurred between 2002 and 2015 that may be attributable to more effective treatment including smoking cessation.

It does not look at the basis of diagnosis as only 64% of patients had spirometry performed prior to baseline.

Comparisons would be more meaningful if they were between subjects with similar levels of impairment.

Author Response

Comments: This is a non-medical purely economic assessment of COPD in Italy. It provides no reference to the reasons for the increase in medical costs or to the trends in tobacco smoking that underlie the occurrence and prognosis of the disease. It does not consider the change in prognosis that occurred between 2002 and 2015 that may be attributable to more effective treatment including smoking cessation. It does not look at the basis of diagnosis as only 64% of patients had spirometry performed prior to baseline. Comparisons would be more meaningful if they were between subjects with similar levels of impairment.

Response: The Italian pharmacoeconomic studies summarized and discussed in the present paper were carried out in population samples comparable in terms of COPD severity (% of mild, moderate, severe), age. gender, and smoking habit (% of active, and ex smokers).

This was the main strenght of these studies to compare in ordere to obtain a reliable trend in cost over the last two decades.

The main messagesa emerging are: 1) the under-esteem and the still insufficient management and governance of COPD in Italy, 2) the too large use of hospitalization, thus mirroring the insufficient role of territorial role of GPs in COPD governance; 3) the progressive incrase of cost is mainly due to hospitalization and inappropriate or inssuficent pharmacological and non-fharmacological (such as rehabilitation) tretamens; 4) the use of annual COPD cost is the best global predictor of moprtality because contains and summarizes all the different conponenents of COPD governance (or non governance) in real life; 5) when condìsidering the high difficulties for defining COPD oprognosis and mortality of Italian patients, at present the annual cost should be regarded as the easiest and less expensive predictor of COPD mortality in clinical practice; 6) the appropriate use of respiratory drugs consents substantial savings and a substantial drop of overall COPD annual cost already in the mid-term, even if the sole cost for drugs is increasing. Finally, health care decisionmakers should pay much more attention to health care costs in order to plan much more effective programd of COPD contaiment.

These are the only messages emerging from the present review of Italian cost of COPD and to emphasize.  

Round 2

Reviewer 1 Report

This paper reflects the comments and corrections of reviewer. Nevertheless, the definition and meaning of 'PRO-BODE score' are unclear and it is difficult to interpret the estimation results. A description of the 'PRO-BODE Index' should be added to improve readability.

Author Response

This paper reflects the comments and corrections of reviewer. Nevertheless, the definition and meaning of 'PRO-BODE score' are unclear and it is difficult to interpret the estimation results. A description of the 'PRO-BODE Index' should be added to improve readability.

Response: Firstly, thakyou very much indeed for having valued the paper by 4 stars. 

As concerning the better interpretation of the PRO-BODE score, it would be noted that it takes origin from the implementation of the widely known BODE index (previously regarded as the most effective indicator for predicting COPD mortality, and based on BMI and difefrenta spects of lung function) with the cost of the two main annual components of COPD clinical progression, such as the number of annual exacerbations and hospitalizations. This new score (PRO-BODE Index) allowed to improve the strenght of relationship  with mortality at three years from r=021 (with the BODE index) to r=0.65 (with the PRO-BODE index: thus emphasizing the relative high role of annual cost assessing for predicting COPD mortality. Higher the annual cost, shorter the survival, and annual cost is much easier to calculate annually.

Reviewer 2 Report

Some reference to the need to control of tobacco smoking as a cause and management issue for COPD is needed. As a "health strategy" it would be at least as important as air pollution due to industrial emissions. It would be relevant to indicate how the prevalence of smoking is changing: in many countries it is declining and more resources are being put into treating nicotine dependence. 

Author Response

Some reference to the need to control of tobacco smoking as a cause and management issue for COPD is needed. As a "health strategy" it would be at least as important as air pollution due to industrial emissions. It would be relevant to indicate how the prevalence of smoking is changing: in many countries it is declining and more resources are being put into treating nicotine dependence.

Response: Firstly, thank you for generally valuing the paper with 4 stars.

As concerning your last request, the meaning of your comments was already included in the text, in particilar in line 183, and in lines 190-194. Actaully, the paper was oriented to only evaluate the changes in costs of COPD over a long period of time.

Anyway, a specific line concerning your request has been implemente in the abstract (lines 22-23, in blue):  " ..., even if the tobacco smoke was progressively declining up to 21% in general population..."